# Dog visits in nursing homes – increase complexity or keep it simple? A randomised controlled study

**Karen Thodberg**[1]*, **Poul B. Videbech**[2], **Tia G. B. Hansen**[3], **Anne Bak Pedersen**[1], **Janne W. Christensen**[1]

1 Department of Animal Science, Aarhus University, Aarhus, Denmark, 2 Psychiatric Centre Glostrup, Glostrup, Denmark, 3 Center for Developmental and Applied Psychological Science, Aalborg University, Aalborg, Denmark

* karen.thodberg@anis.au.dk

## Abstract

### Objective

To compare the immediate response of nursing home residents to dog visits with or without an activity, and the impact of cognitive ability.

### Methods

In a randomly controlled trial, 174 nursing home residents were allocated to 12 bi-weekly 10-minute visits: either ordinary dog visits (D, n = 57, 49 analysed), dog visits with an activity (DA, n = 56, 48 analysed), or visits with activity but no dog (A, n = 61, 54 analysed). We recorded frequency and duration of residents' verbal and physical interactions with the dog and persons. Data were analysed in three periods of four visits (period 1–3) as binomial variables (generalised linear models) or durations (non-parametric statistics).

### Results

Both visit type and impairment level affected the likelihood of interacting with the dog (D and DA). In some periods increased cognitive impairment lowered odds of touching the dog in DA visits (period 1: $F_{1,85} = 5.17$, P < 0.05) and talking to it directly (period 1: $F_{1,90} = 4.60$, P < 0.05; period 3: $F_{1,87} = 5.34$, P < 0.05). Throughout, residents talked less to persons during DA visits compared to D and A (P = 0.01–0.05), and level of cognitive impairment correlated negatively with talk duration (P < 0.001). Generally, high cognitive impairment level lowered odds of interacting with (period 1: $F_{1,89} = 7.89$, P < 0.01; period 2: $F_{1,97} = 6.76$, P = 0.01; period 3: $F_{1,92} = 13.57$, P < 0.001) and talking about the activities (period 1: $F_{1,89} = 13.78$, P <0.001; period 2: $F_{1,88} = 3.27$, P = 0.07; period 3: $F_{1,86} = 3.88$, P = 0.05).

### Conclusion

Visits without specific activities stimulated residents to interact with the dog, whereas increasing the complexity of dog visits by adding activities resulted in less interaction with the dog for severely impaired residents. The optimal dog visit for the less cognitively impaired residents

**Data Availability Statement:** We are unable to make an anonymized dataset publicly available according to Danish law, as the dataset contains both sensitive information and information which

can be used to identify the study participants. These rules are based on the Data Protection Act, imposed by The Danish Data Protection Agency. An English translation of the Data Protection Act can be found on the official website for The Danish Data Protection Agency (https://www.datatilsynet.dk/english/legislation/). Data are available from the Danish National Archives (https://www.sa.dk/en/k/about-us) upon request and they can be contacted by email at dt@datatilsynet.dk. In order to use the data in research there must be a contract and a further approval from The Danish Data Protection Agency. For more information on how to acquire the data, please contact Karen Thodberg (karen.thodberg@anis.au.dk). Following information will be required at the time of application: a description of how the data will be used, securely managed, and permanently deleted.

**Funding:** The study was supported by a grant from TrygFonden to KT (106330; TrygFonden.dk). The funders had no role in study design, data collection and analysis, decision to publish, or preparation of the manuscript.

**Competing interests:** The authors have declared that no competing interests exist.

could include activities and thereby a possibility to interact with the dog in different ways, whereas for severely impaired residents, just being with the dog seems more appropriate.

## Introduction

Dog visits have become a popular activity in many nursing homes in Europe [1, 2], and organisations in Denmark and several other countries recruit volunteers and their dogs in order to facilitate contact to animals for nursing home residents [3, 4].

Studies of how visits from dogs affect nursing home residents have mainly focussed on long-term benefits measured by psychometric scales evaluating symptoms of depression, loneliness, quality of life, impairment level and agitated behaviour in nursing home residents or persons with dementia. Recently, review papers have outnumbered papers reporting experimental studies, and authors generally agree that the evidence base for specific effects of dog visits in nursing homes is weak and inconclusive [5–10]. Several reviews point out that the use of very different study settings makes it difficult to reach definite conclusions about specific effects of dog visits [6–8, 11–13]. These differences in study setup include the dogs' role during visits, duration and frequency of visits, a large variation in the studies' perspective and focus, difference in study design, not to mention suboptimal study designs, such as lack of appropriate control groups. Several reviews conclude that implementing dog visits in nursing homes has the potential to be a supplement treatment or activity and that future studies should investigate the actual mechanisms of how dogs affect nursing home residents while using appropriate effect measures of this [12–14].

How the residents interact with dogs during visits could be the key to a better understanding of potential long-term effects [15–17]. In a previous study, we found that individual visits to nursing home residents accompanied by a dog, compared to a soft toy animal, stimulate the residents to engage more in physical interaction, conversation and eye contact to the visiting animals [18]. Visits, where an animal-like alternative–the robotic seal Paro–was brought along, also stimulate residents to touch, to make conversation and establish eye contact, but compared to the stimulation of the real dog it was less persistent over a series of 12 visits, presumably because a real dog provide more means of interaction and feedback. A few other studies of dog visits to groups of residents also found that the presence of a dog during visits stimulates social interaction, but the group settings used in these studies make it difficult to isolate the specific effect of dog contact [19–22].

Nursing home residents comprise elderly persons with a variety of physical and cognitive disabilities and functional impairments. In general, being engaged in everyday activities seems to increase well-being in nursing home residents [23, 24], but residents with severe dementia generally participate less in activities [25, 26]. Thus, the heterogeneity of the population raises the question as to how to make suitable activities, which both fulfil different needs and wishes, and comply with the wide range in cognitive impairment level, and the ability to move, interact and communicate? Dog visits and interacting with dogs could be such an activity, but we need more knowledge regarding the applicability and benefit for nursing home residents with varying levels of cognitive disabilities and functional impairments.

In our previous study, we found that more severely impaired persons prefer to interact with an animal–dog, robot seal or toy cat–instead of a human during visits, whereas the opposite was found for the more well-functioning residents, probably due to the more non-demanding nature of the non-human interaction [18]. Only two other studies have looked at whether cognitive impairment affects nursing home residents' immediate response to dog visits [20, 22].

One study of dog visits in a group setting found that participants living in nursing homes, where the majority were assessed to have severe dementia, look less at activities with the dog and sleep more during the interventions, compared to participants who lived on their own and suffered from moderate or mild dementia [20]. Friedman et al. (2019) [22] report no association between impairment level and recorded behaviour during dog visits in groups.

Dog visits offer an opportunity to introduce a range of activities, including activities that do not solely rely on human-human interaction and communication, and inviting residents to take part in specific activities together with a dog could potentially increase the level of interaction. However, introducing extra activities during dog visits will also make visits more demanding for the residents because they may feel a pressure to participate in the activities and to perform in a certain way. This increase in visit complexity could perhaps be especially challenging for cognitively impaired persons.

Thus, adding an activity to a dog visit will most likely affect how the resident acts and responds–for some it might be an improvement, but maybe not for all, depending on their cognitive abilities, general condition and other handicaps.

We aimed to study whether enhancing the possibility for interaction with the dog during dog visits is beneficial for all, as measured by their immediate behavioural response. This was achieved through the following partial aims:

1. To study whether introducing an activity with a dog during bi-weekly dog-assisted visits would affect the nursing home residents' behaviour during visits, and especially their level of interaction with the dog.

2. To study how the nursing home residents' level of cognitive impairment affected how much they interacted with the dog, the persons present during the visits, and their engagement in an activity.

3. To study how resident characteristics, such as age, sex and physical ability affected their behaviour during the visits.

We used a randomised and controlled study design, and to control for the effect of adding an activity to the dog visits we compared the following three visit types: 1) Visits with a dog, and no additional activity, 2) Visit with a dog, with an additional dog-related activity or 3) Visits without dog, but with an additional activity.

Our hypotheses were that: a) adding an activity to a dog visit would increase interactions with the dog, b) the extra complexity and stimulation provided by adding an activity to dog visits would motivate less impaired residents to participate more actively in the visits c) the extra stimulation of adding an activity to dog visits could overstimulate residents with severe cognitive impairment and therefore not necessarily enhance their engagement in the visits, d) the residents' potential physical disabilities would limit their engagement in the visits in general and the activities in particular.

The visits were conducted individually with one resident at the time in order to separate the effect of the visit itself from the effect of being in a group of residents.

The study was part of a larger study on immediate and long-term effects of bi-weekly dog visits with or without added activity, and the results on long-term effects will be presented elsewhere.

## Materials and methods

### Participants, consent and approval of study

A total of 186 nursing home residents from seven nursing homes in Denmark were enrolled in the study. In Denmark, the average age when moving into a nursing home is 83.7 years, and

approximately 42% of nursing home residents are diagnosed with dementia, but the precise prevalence of dementia is unknown [27]. Written informed consent was obtained from all participants or their relatives. The participants could withdraw from the study at any point. Inclusion criteria were that the participants were able to sit up during visits. Exclusion criteria were allergic reactions or fear of dogs. None of the residents had pets of their own, but in a few cases the nursing homes had a resident cat. The study was approved by *The Scientific Ethical Committee for Denmark*. The clinical trial was not registered before conducting the study, as we were not aware that this was the general practice for this type of study. However, the study is now registered, with the registry name "Optimal Dog visits" and the ID number "NCT04635124" (https://clinicaltrials.gov/ct2/show/NCT04635124). Data were collected from August 2015 to October 2017. Fig 1 shows the CONSORT flowchart.

## Design

The design of the study was a stratified and randomized complete block design. Each nursing home was a block. After informed consent had been given, the participants were allocated randomly to one of the three visit types, while ensuring that the participants' level of cognitive impairment was balanced between visit types. This was achieved by ranking the participants according to MMSE score within sex and allocating them to the three visit types by using different numerical orders in each nursing home (1, 2, 3; 3, 1, 2; 2, 3, 1 etc.). The numerical orders used were allocated randomly to the seven nursing homes by the principal investigator. This procedure for randomisation deviates from the registered protocol. We originally planned to divide the participants into two groups based on their level of cognitive impairment but modified this due to lack of information regarding the distribution. The behaviour variables are primary outcome measures in this study and the sample size was based on data from [18], where physical contact to the dog, and talking about the/visit object were also recorded. The difference in mean durations (± standard deviation) in the experimental treatment and a control treatment (118 vs 51, standard deviation = 110 and 36 vs 10, standard deviation = 34) were used as effect sizes in the power analysis. The required sample sizes for obtaining a power of 80%, assuming a statistical significance of 0.05 (two-sided), were 44 and 28, for these variables. We used the largest estimate to decide the sample size, taking into account that we could expect dropouts (20% of the recruited participants in [18]), and therefore aimed at recruiting 56 participants for each visit treatment (n = 24 in each nursing home), adding up to a total sample size of at least 168 recruited participants.

## The intervention and the visit types

The residents were visited individually and, apart from the visitor, an observer was also present, making direct recordings of the resident's behaviour. Each resident was scheduled to receive two visits per week, either Mondays and Wednesdays or Tuesdays and Thursdays for 6 weeks, i.e. a total of 12 visits. The visits were conducted between 9 am and 4 pm, and within this timeframe the visits were planned to suit the circadian rhythm of the individual participant. In each nursing home, there were two visitors and two observers (project staff), and they visited an equal number of residents in each visit type. The residents met the same visitor and dog (if allocated to receive dog visits) in each visit, whereas the two observers alternated to ensure standardisation of the visits between the two visitors. Each visitor worked with a specific dog; hence, two dogs participated in each nursing home.

The visits took place in the residents' own small apartment in the nursing home, except for few exceptions where the visit took place in another undisturbed area. Usually, the resident

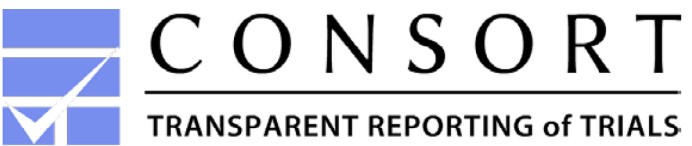

## CONSORT 2010 Flow Diagram

**Enrollment**

Assessed for eligibility (N=186)

Excluded  (n=1)
♦ Declined to participate (n= 10)
♦ Died (n=2)

Randomized (N=174)

**Allocation**

Allocated to visit type **D** (n=57)
♦ Received allocated intervention (n=49)
♦ Did not receive allocated intervention (revoked consent (n=6); less than 6 visits (n=1) died before having 6 visits (n=1)

Allocated to visit type **DA** (n=56)
♦ Received allocated intervention (n=48)
♦ Did not receive allocated intervention (revoked consent (n=5); less than 6 visits (n=2) died before having 6 visits (n=1)

Allocated to visit type **A** (n=61)
♦ Received allocated intervention (n=54)
♦ Did not receive allocated intervention (revoked consent (n=3; less than 6 visits (n=3); died before having 6 visits (n=1)

**Analysis**

Analysed, visit type D (n=49)
♦ Excluded from analysis (n=0)

Analysed, visit type DA (n=48)
♦ Excluded from analysis (n=0)

Analysed, visit type A (n=54)
♦ Excluded from analysis (n=0)

**Fig 1. CONSORT flowchart.**

would sit in a normal chair or a wheel chair during the visit. The participants were free to move around during the visit, but this rarely happened.

The visits lasted 10 minutes, excluding the time spent entering and greeting as well as closing the visit and leaving the room.

The residents' 12 visits were either 1) Visits with a dog, no activity (D), 2) Visits with a dog, with activity (DA), or 3) Visits without dog, with an activity (A). The residents that received dog visits (D, DA) met the same dog in all 12 visits. In all visit types, the visitor positioned herself close by the resident. The observer placed herself further away to be able to overlook the situation, but joined the conversation when it felt natural.

Visits with a dog but no activity (D) followed the same schedule as the dog visits in Thodberg et al. (2016). To ensure that the resident could touch the dog during the visit, it was within reach for at least 80% of the time, unless the resident clearly showed no interest at all or disapproved of contact with the animal. The dog was on leash. During the first visit, we introduced the dog and encouraged the resident to take contact. To start the conversation, the subject "dog" was chosen, unless the resident changed the subject. The first six minutes of the visit, we incited the resident to focus on the dog, but the remaining four minutes, contact to the dog was on the resident's own initiative.

In visits with a dog and an activity (DA), the first two minutes were identical with D. After two minutes, the visitor introduced an activity for the resident that involved interaction with the dog, e.g. brushing the dog or hiding treats for it to find. A new activity was introduced to the residents in each of the 12 visits (Table 1), and the order of activities was randomized across residents. In the first four minutes after introducing the activity, the visitor incited the resident to take part in the activity, unless he or she directly or indirectly expressed that it was of no interest. In the remaining four minutes of the visit, the initiative to take part in the

**Table 1. Description of the 12 different activities used in visit type DA.**

| | Activity | Description |
|---|---|---|
| 1 | Brushing dog | Hold the brush and brush the dog |
| 2 | Hide treats in cup (cups) | Choose one out of five cups and hide a treat in it. Place the cups upside down on the floor for the dog to choose. |
| 3 | Treats in a bottle | Hold the plastic bottle and put treats in it. Place the bottle on the floor and watch the dog try to get the treats out. |
| 4 | Hiding treats in the room | Hide three treats in the resident's room for the dog to find. (The residents choose where and maybe help to hide treats). Start the stopwatch and time how fast the dog finds the treats. |
| 5 | Playing dice about training tasks | Play dice to decide which commands to give to the dog (1: sit, 2: give paw, 3: stand, 4: lie down, 5: bark, 6: high five). |
| 6 | Compare the dog's coat with fur/pelt from another animal | Touch and compare the dog's coat and pelts from other animals. |
| 7 | Hiding treats in a ball | Hold the rubber ball and put treats in it. Place the ball on the floor and watch the dog try to get the treats out. |
| 8 | Pretending that the dog is reading | The resident shows the dog signs with written commands. We signal the commands to the dog (hand signals), so that it seems that the dog is reading the signs. After some repetitions, we reveal that we are helping the dog, and optionally the resident can try the hand signals |
| 9 | Training the dog | Deciding what the dog should do, and give the commands |
| 10 | Dog game 1 (commercial dog game) | Hide treats and time how fast the dog finds all the treats. |
| 11 | Dog game 2 (commercial dog game) | Hide treats and time how fast the dog finds all the treats. |
| 12 | Brush dog and compare the dog's coat with pictures of different dog breeds | Brush the dog and identify similarities and differences between the visiting and the dogs in the pictures. |

DA: visits with a dog and an activity.

activity and interact with the dog was left to the resident. The availability of contact to the dog was as in visit type D, but during some activities the dog was unleashed if necessary for it to take part in the activity.

In visits without a dog, but with an activity (A), the visitor also introduced the activity after two minutes. As in DA visits, there was a new activity for each of the 12 visits (Table 2), presented in random order to the individual residents. As in DA visits, the visitor incited the resident to take part in the first four minutes after introducing the activity, and then left the initiative to the resident in the remaining four minutes of the visit.

The dogs were all approved by *TrygFonden*, a Danish organization that certifies family dogs and their owners to work as volunteers with dog-assisted interventions in nursing homes [3] in accordance with the guidelines for human and animal welfare in Animal Assisted Activities suggested by IAHAIO, which is a global network of HAI organisations [28]. The six dogs were approximately the same size, and were either retrievers or retriever mixes (four males (5–8 years) and two females (6–8 years)). All dog owners had liability insurance for their dog. The visitors were not the owners of the dogs, but all spent time with the dogs before the intervention periods started to get to know the dog, and vice versa, and try out the different activities for the DA visits. It was our first priority that the dogs were relaxed with the handler and the visits, and less important that they could do the tasks related to the activities perfectly. The dogs were carefully looked after throughout the study, and the handlers were especially observant with regard to participants' behaviour towards the dogs. No situations occurred where the welfare of the dogs was considered at risk.

**Table 2.  Description of the 12 different activities used in visit type A.**

| Activity | | Description |
| --- | --- | --- |
| 1 | "Memory" with proverbs | Read from cards with the first line of a proverb and then finish it. |
| 2 | "Memory" with songs | Read from cards with the first line of a song and then finish it |
| 3 | Pictures of royal persons | Look at and talk about pictures of previous kings and the present royal family |
| 4 | Pictures of cars/clothing from different decades | Look at and talk about pictures of new and old cars, or clothing from different decades |
| 5 | Pictures of people from different cultures | Look at and talk about pictures of people from different cultures |
| 6 | Reading and looking at the newspaper of the day | Read and look at the newspaper of the day. Have a conversation about the headlines and topics |
| 7 | Listening to music | Listen to different types of music, talk about it and maybe sing along |
| 8 | Touching pelts | Touch and talk about pelts from different animals. Conversation about from which animals the pelts come from. |
| 9 | Looking at and sniffing at things from the nature | Look at and sniff at things from the nature. Conversation about what it is. The things varied according to the season |
| 10 | Looking at objects from the Second World War (pictures, ration coupons, resistance emblems) | Explore the things and have a conversation about them |
| 11 | Looking at school things from the old days | Explore the things and have a conversation about them |
| 12 | Looking at working and kitchen tools from the old days | Explore the things and have a conversation about them |

A: visits with no dog and with an activity.

## Description of the population

Basic demographic information (age, sex, time lived in the nursing home, civil/marital status), information related to the residents' state of health (current psychiatric disorders, e.g. dementia, Down's syndrome, schizophrenia, bipolar disorder), physical handicaps (impaired vision, impaired hearing, impaired speech ability, paralysis) and time spent in bed (hours bed bound/day), were provided by the nursing home staff as approved by *The Scientific Ethical Committee for Denmark*.

In the week preceding the intervention period, the participants were interviewed by a project nurse, and two psychiatric measurements were used: 1) Mini-Mental State Examination (MMSE), which gives information about the participants' cognitive status [29]; 2) The Gottfries-Bråne-Steen Scale (GBS), which is an evaluation of disabilities, language, psychiatric symptoms, average daily living function and behaviour of the participants [30]. The values measured prior to the intervention period are reported here, and the measures taken after the experimental period will be published elsewhere.

If participants were unable to answer specific items from the psychiatric scales, due to impairment of vision or hearing, or due to general reluctance, their answer was marked with a specific score. This was done to be able to distinguish between answers reflecting the mental state measured by the scale from reluctance to answer or for example not being able to hear the questions asked. If residents were unable or reluctant to reply in more than 50% of the items of the scale, they were excluded from the statistical analyses. For those with less than 50% missed items, the values of the missed items were substituted with the median of this item across participants by imputation [31]. The measures of GBS and MMSE were highly correlated and as the GBS score had a much lower level of imputation (2%), compared to the MMSE score (25%), we used GBS as the explanatory variable for cognitive impairment level instead of MMSE in the statistical analyses.

## Behavioural observations

The behaviour of the residents was recorded by direct observation using continuous recording [32]. In all visits, the frequency and duration of talk and physical contact to the persons present were recorded. Talk was classified into talk about the dog and/or activity, depending on the visit type. In visits with dogs (D, DA), the frequency and duration of physical contact and talk directed to the dog were recorded, and in visits where we introduced an activity (DA, A), the frequency and duration of the residents' involvement in the activity were recorded (Fig 2). The observer recorded behaviour during the visits using a tablet with a touch screen, as described in Thodberg et al. (2016) [18].

## Statistics

All data were analysed using SAS 9.3.

**Behaviour variables.**   From the variables describing the residents' physical contact with the dog (Fig 2) we calculated one composite variable, *physical contact to dog. Physical contact to activity* comprised actually using the remedies related to the activity as well as just touching them. Conversation was analysed as *talk in total, talk directed at the dog* and as *talk directed at a person. Talk about the dog* and *talk about the activity*, which were subsets of *talk directed at a person*, were also analysed separately (Fig 2).

For the statistical analysis, the 12 visits were divided into three periods, with each period covering four visits (period 1: visits 1–4, period 2: visits 5–8, period 3: visits 9–12). The variables used in the analyses were the mean of a given variable during the visits within periods. If the residents had received fewer than six visits in total, e.g. due to illness or dropout, they were

| Behaviour categories and variables | Visit type | | | Description | Variable/ Composite variable | | |
|---|---|---|---|---|---|---|---|
| | D | DA | A | | | | |
| **Physical contact** | | | | | | | |
| To dog, active*** | x | x | | Intentionally touching the dog with one or both hands, or other parts of the body (e.g. foot, leg, head). Also scored if the resident engages in the activity or touches the remedies of the activity at the same time during DA visits. | **Physical contact to dog**\*\* | | |
| To dog, passive*** | x | x | | Unintentional touch, e.g. because the dog leans against the resident or is lying on his or her feet. Also scored if the resident engages in the activity or touches the remedies of the activity at the same time during DA visits. | | | |
| To person**** | x | x | x | Intentionally touching a person with one or both hands, or other parts of the body (e.g. foot, leg, head). | | | |
| Touching and using things or remedies related to the activity*** | | x | x | Using the remedies of the activity, engaging in the activity as intended or examining the remedies with the fingers. Also scored if the resident touches the dog at the same time during DA visits. | **Physical contact to activity**\*\*\* | | |
| Passive contact to things or remedies related to the activity*** | | x | x | Holding the remedies of the activity without using them or not engaging in the activity as intended. Also scored if the resident touches the dog at the same time during DA visits. | | | |
| **Talking** | | | | | | | |
| **Talk directed at the dog**\* | x | x | | Talking directly to the dog | *Talk in total* | *Talk directed at a person* | **Talk about the dog**\*\* |
| Talk to person about the dog*** | x | x | | Talking directly to the visitor or the observer about the dog, but in DA visits not when in relation to the activity | | | |
| **To person, about the activity**\*\* | | x | x | Talking directly to the visitor or the observer about the activity, in DA visits this includes the dog's role in the activity | | | |
| To person, about other things*** | x | x | x | Talking directly to the visitor or the observer about other things than the dog or the activity | | | |
| Non-verbal utterance**** | x | x | x | Talking that is either unintelligible or with no verbal content | | | |

**Fig 2. Ethogram, composite variables and use of statistical models.** The column "Variable/Composite variable", shows the components of the composite variables. The use of statistical models is indicated with bold text for the generalised linear mixed model (Likelihood of occurrence, in bold), italics for non-parametric analysis (duration, in italics). The cut-off for the binary variables in the generalised linear mixed models could be either 0 or 60 seconds, indicated by * and **, respectively. ***Not analysed separately, ****Not analysed.

not included in the statistical analyses. The behaviour variables were all analysed for the total visit duration (10 minutes), and additionally, some variables were analysed for the first two minutes of the visits.

**Parametric models.** Behavioural variables were analysed using a generalised linear mixed model and entered as binomially distributed, with a logit link function (Logistic regression; PROC Glimmix). For some variables, the cut-off for the binary variable was 'occurrence vs non-occurrence', that is whether the behaviour occurred for more than 0 seconds or not at all, whereas for other variables the cut-off was 'occurrence for more than 60 seconds or less' (Fig 2). The cut-off was based on the distribution of the variables. The categorised explanatory variables (possible levels) were visit type (D, DA, A), sex (male, female), identity of visitor (6), identity of observer (4), identity of dog (6), nursing home (7), ability to hear (normal, impaired), speech ability (normal, impaired), vision (normal, reduced), and mobility (normal, or at least partly paralysed). Continuous explanatory variables (range) were age (61–100), initial GBS score of the residents (5–120) and the total number of experimental visits received during the intervention period (6–12). The only interaction included in the model was between visit type and the initial GBS-score. We reduced the full model by backwards reduction, and factors were removed when P-values exceeded 0.05, except for visit type, which was retained. The appropriateness of the final models was evaluated by using the dispersion parameter (Pearson chi-square/DF), which optimally should be close to a value of one, and ranged from 0.91 to 1.07 in the final models. Results are presented as odds ratios (LSMEANS with 95% confidence intervals) and in some cases as predicted probabilities. In cases where the final

models include a significant interaction between the visit treatment and the GBS score (continuous), we present the odds as well as the predicted probabilities estimated by the model for the GBS values 18, 48, and 98, representing the actual range of scores in the study population.

**Non-parametric models.** Variables that were not normally distributed or could not be analysed with logistic regression were analysed non-parametrically. Differences in baseline measures between treatment groups (visits types) and nursing homes were analysed by the *Kruskal-Wallis Test*, and for pair-wise comparisons *Wilcoxon Rank Sum Test* was applied (Both using Proc Npar1way).

The same two tests were used to analyse the effect of visit type on *physical contact to persons*, *talk in total and talk directed at a person* within each period. The development over periods was analysed with Wilcoxon Signed Ranks Test (Proc Npar1way) for pairwise comparisons of repeated measures.

To analyse time development in the behaviour variables analysed by logistic regression, we used the estimated probabilities of the behavioural response variables [33] from the final logistic regression models for each period as dependent variables in the Wilcoxon Signed Ranks Test for pairwise comparisons of repeated measures. Using this method, the dependent behaviour variable was already corrected for the significant effects found in the logistic regression models, and provides a more clear picture of potential effects of period, compared to if we had used the raw variables as a dependent variable.

Fishers Exact test (Proc Freq) was used to analyse whether residents that dropped out or had less than six visits were equally distributed over visit types.

Results of the non-parametric analyses are presented as medians (± IQ-range).

To describe the association between cognitive impairment and the variables *talk directed at persons* and *talk in total* we used Spearman rank-order correlation coefficients which were presented as $r_s$ with a P-value and N (number of observations; Proc Corr).

## Results

### Description of participants

Of the 186 participants enrolled in the study, two died, and 10 revoked their consent to participate before the experimental visits began, resulting in 174 participants at the beginning of the intervention period. Of these, six died during the intervention, and of the remaining 168 participants 15 withdrew during the intervention period, either due to illness or because they did not want to receive more visits, and six participants had less than six visits due to different reasons. We analysed data from 151 participants as one of the participants that dropped out, and three of the persons that died during the intervention period, had more than six visits and were therefore included (Fig 1). The 23 participants that either left the study or had less than six visits were equally distributed on visit types (D: 8, DA: 8, A: 7; P = 0.84). The participants' mean age was 86 years, and they had lived, on average, 42.3 months in the nursing home when they entered the study. The majority were women (64.2%) and the mean time spent in bed per 24 hours was 12.3 hours (Table 3). The majority were widowed (63.6%), 21.9% were married, and the remaining were either unmarried (5.3%) or divorced (6.6%; NA: 2.7%). Based on the staff's information, approximately one third of the participants had impaired hearing (32%; NA: 4%) and reduced eyesight (34%; NA: 6%), 12% were partly paralysed, ranging from one limb being impaired to being paralysed in most of the body (NA: 1%), and 18% were speech impaired (NA: 2%).

The median scores of the psychiatric scales measured before the intervention period are shown in Table 4, and none of them differed significantly between treatment groups or nursing homes. Just above 40% of the participants (40.5%) were diagnosed with dementia, and

**Table 3. Description of the study population.**

| Nursing home | Age (years) | Time lived in the nursing home (months) | Percentage of women (%) | Time in bed (h per 24 h) | Number of visits received | Number of participants in each visit type (D, DA, A) |
|---|---|---|---|---|---|---|
| 1 (n = 20) | 84.3 | 50.8 | 65.0 | 12.9 | 10.0 | (7, 8, 5) |
| | 84 (73–100) | 33 (3–229) | | 11.8 (9–21.5) | 11 (6–12) | |
| 2 (n = 25) | 85.0 | 33.9 | 64.0 | 11.9 | 10.6 | (9, 7, 9) |
| | 85 (69–97) | 28 (3–76) | | 12 (8–19.5) | 11 (6–12) | |
| 3 (n = 14) | 85.6 | 27.5 | 57.1 | 13.2 | 11.1 | (4, 5, 5) |
| | 87 (61–100) | 18.5 (0.5–88) | | 12 (9–22) | 11 (10–12) | |
| 4 (n = 24) | 82.6 | 56.8 | 54.2 | 12.2 | 10.5 | (7, 7, 10) |
| | 83.5 (64–95) | 27 (1–187) | | 11 (4–22) | 11 (7–12) | |
| 5 (n = 21) | 84.3 | 51.9 | 61,9 | 12.2 | 10.4 | (7, 6, 8) |
| | 87 (70–97) | 49 (5–185) | | 11.5 (8–20) | 10 (6–12) | |
| 6 (n = 24) | 85.3 | 45.9 | 66.7 | 10.4 | 11.0 | (10, 6, 8) |
| | 87.5 (67–96) | 24 (6–191) | | 10 (0–18) | 11 (7–12) | |
| 7 (n = 23) | 86.0 | 26.9 | 78.3 | 13.2 | 10.2 | (5, 9, 9) |
| | 87 (73–97) | 15 (1–125) | | 11 (7–24) | 11 (6–12) | |
| Total (N = 151) | 84.7 | 42.3 | 64.2 | 12.2 | 10.5 | (49, 48, 54) |
| | 86 (61–100) | 28 (0.5–229) | | 11 (0–24) | 11 (6–12) | |

The age of participants, time lived in the nursing home, time in bed per 24 h and number of visits are given as means and medians with interquartile range.

16.6% had another psychiatric disorder, whereas 35.8% had no psychiatric diagnoses (NA: 7.3%).

## Behaviour recorded in all visit types: *Talk in total*, *Talk directed at persons* and *Physical contact to persons*

**Entire visit.** In all three periods, the residents directed less talk to persons (visitor or observer) if they received dog visits with an activity (DA) compared to the other visit types (P = 0.01–0.05 in all periods; Fig 3). Throughout, the more cognitively impaired residents talked less with the person present, compared to less impaired residents, as shown by the negative correlations between the GBS score and duration of talking to persons (period 1: $r_s$ =

**Table 4. Psychiatric measures before the six-week intervention period.**

| Nursing Home | MMSE (n) | GBS (n) | MMSE | GBS |
|---|---|---|---|---|
| 1 | 13 | 19 | 14 [6; 19] | 51 [32; 68] |
| 2 | 19 | 25 | 18 [10; 22] | 40 [28; 63] |
| 3 | 10 | 13 | 17.5 [13; 22] | 43 [24; 54] |
| 4 | 18 | 24 | 13 [6; 19] | 58 [18.5; 84] |
| 5 | 17 | 21 | 20 [10; 23] | 35 [22; 46] |
| 6 | 22 | 24 | 15 [8; 21] | 56.5 [34; 67.5] |
| 7 | 15 | 22 | 14 [10; 18] | 59.5 [31; 85] |
| Total N = 114 | 114 | 148 | 15.5 [9; 21] | 46 [26.5; 64] |

None of the scores differed between nursing homes or treatment groups.

The scores are shown as medians and interquartile ranges, as well as the number of applicable observations in each nursing home.

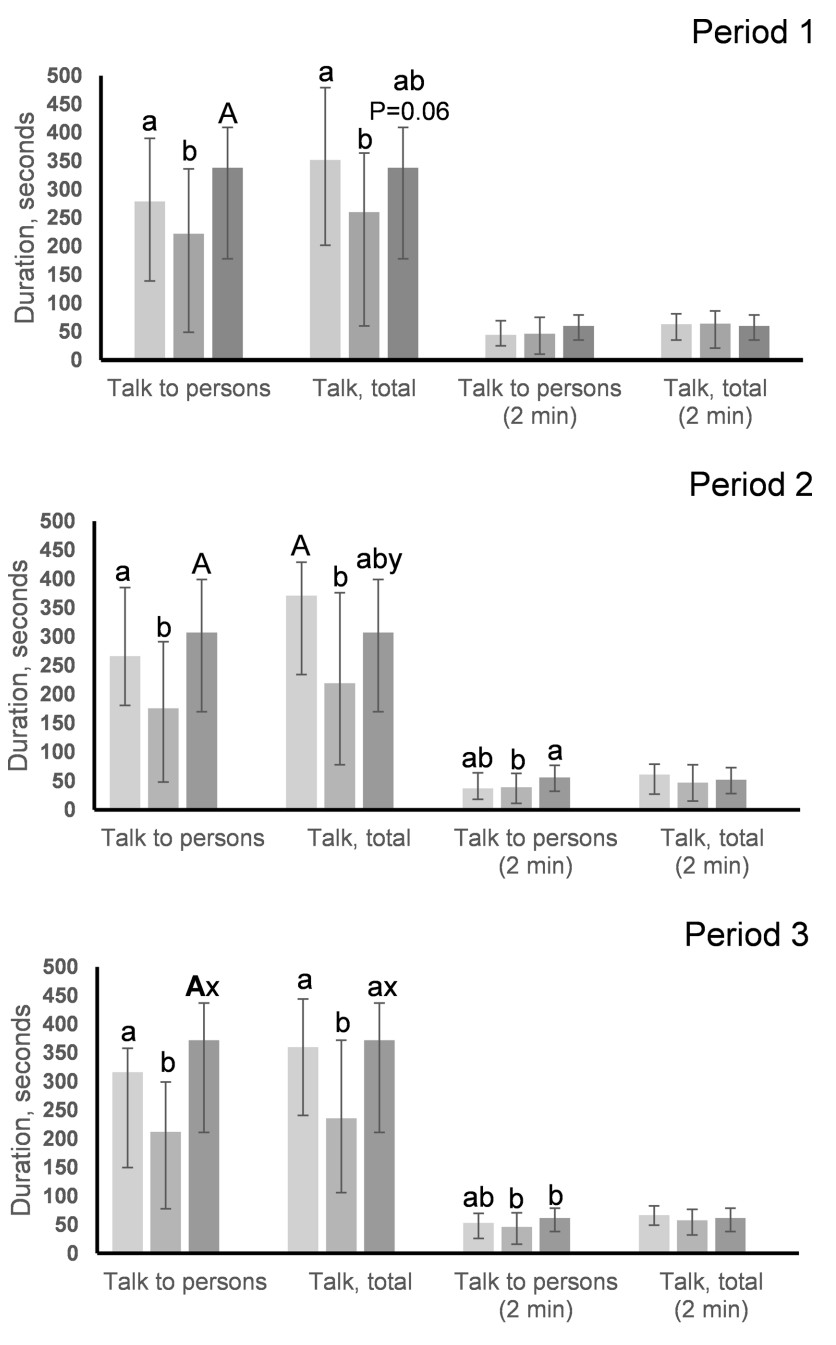

**Fig 3. Duration of *talking to person* and *talking in total* for the entire visit and in the first two minutes in period 1, 2 and 3.** Results from the entire visit and in the first two minutes in periods 1, 2 and 3 are shown as the median in seconds and interquartile range (IQ). Differences between visits within period are indicated with the letters a and b when P < 0.05, with A and B for P < 0.01, and **A** and **B** for P < 0.001. Differences across periods are indicated with the letters x and y for P < 0.05.

-0.36, P < 0.001, N = 148; period 2, $r_s$ = -0.37, P < 0.001, N = 148; period 3: $r_s$ = -0.39, P < 0.001, N = 144). Residents receiving A visits had a higher duration of *talk directed at persons* in the last third of the intervention compared to in period 2 (S = 221.5, n = 52, P < 0.05), but no other differences between periods were found.

*Talk in total* (talk directed both at humans and a dog in D and DA visits) was shorter in DA compared to D visits throughout the intervention (P = 0.01–0.05 in all periods) and shorter in DA visits compared to A visits in period 3 (P = 0.01; Fig 3). For visit type A, *talk in total* is identical to *talk directed at a person*, as there was no dog present in these visits. As for *talk directed at a person*, the more cognitively impaired residents talked less in total compared to less impaired residents (period 1: $r_s$ = -0.39, P < 0.001, N = 148; period 2, $r_s$ = -0.39, P < 0.001, N = 148; period 3: $r_s$ = -0.45, P < 0.001, N = 144). For visit types D and DA, *talk in total* did not differ between periods.

The residents only touched either of the persons a few seconds on average (period 1: 5.4 ± 24.4; period 2: 5.6 ± 22.0; period 3: 6.6 ± 30.4), and this did not differ between visit types in any of the periods.

**Initial 2 minutes of the visit.** In the first two minutes of the visits, we found no differences in the duration *talk directed at a person* in period 1, but in periods 2 and 3 residents receiving A visits talked more to persons than those receiving DA visits (period 2: P < 0.05, period 3: P < 0.05; Fig 3). In visit type DA, the residents talked more to persons in the initial two minutes of the visits in period 1 compared to period 2 (S = -163.5, n = 45, P < 0.05).

Visit type did not affect *talk in total* in the first 2 minutes of the visits (Fig 3), but, in period 2, residents receiving DA visits talked less compared to periods 1 and 3 (period 1 vs 2: S = -175, n = 45, P < 0.05; period 2 vs 3: S = 171, n = 45, P < 0.05).

## Behaviour recorded in visit types D and DA: *Physical contact to the dog, Talk directed at the dog* and *Talk about the dog*

**Entire visit.** In the first third of the intervention, the odds of having *physical contact to the dog* during DA and A visits were affected by visit type and GBS score in interaction ($F_{1,85}$ = 5.17, P < 0.05; Table 5). Increased impairment level increased the odds of touching the dog in D visits, whereas we found the opposite for DA visits (Table 5 and Fig 4). *Physical contact to the dog* was also affected by which person visited the resident ($F_{5,85}$ = 2.48, P < 0.05). In the intermediate part, the GBS score correlated negatively to the likelihood of touching the dog ($F_{1,91}$ = 4.62, P < 0.05), regardless of visit type, meaning that a higher level of cognitive impairment lowered the odds of having physical contact to the dog (Table 5). Throughout the intervention, having many visits in total was related to being more likely to touch the dog (period 1: $F_{1,85}$ = 7.34, P < 0.01; period 2: $F_{1,91}$ = 4.24, P < 0.05; period 3: $F_{1,84}$ = 7.92, P < 0.01; Table 5).

In the first and last third of the intervention period, the likelihood of *talk directed at the dog* was affected by an interaction between visit type and GBS score (period 1: $F_{1,90}$ = 4.60, P < 0.05; period 3: $F_{1,87}$ = 5.34, P < 0.05), showing that severely impaired residents talked less to the dog in DA compared to D visits (Table 5 and Fig 5). In the second and third period of the intervention, the odds of talking to the dog were higher for residents with no speech impairment compared to residents with impaired speech (period 2: $F_{1,90}$ = 11.03, P = 0.001; period 3: $F_{1,87}$ = 3.93, P = 0.05; Table 5). During the intermediate four visits, residents without a physical handicap were 7.7 times more likely to talk to the dog compared to disabled participants ($F_{1,90}$ = 7.68, P < 0.01; Table 5). Both in the first and third period of the intervention, a higher number of total visits were related to talking more with the dog (period 1: $F_{1,90}$ = 6.12, P < 0.05; period 3: $F_{1,87}$ = 10.25, P < 0.01; Table 5).

Halfway through the intervention, residents receiving DA visits had higher odds of *talking about the dog* compared to those having D visits ($F_{1,81}$ = 3.81, P = 0.05; Table 5). However, in the last period, an interaction between visit type and GBS score affected the likelihood of talking about the dog ($F_{1,84}$ = 7.02, P < 0.01, Table 5 and Fig 6), showing that

**Table 5. Results from the analysis of behaviour related to the dog in D and DA visits.**

| **Physical contact to the dog** | | | | |
|---|---|---|---|---|
| *Period 1* | | OR | 95% CI | P-value |
| Interaction | | | | |
| GBS * visit type | D | 1.2[a] | 0.7–2.0 | <0.05 |
| | DA | 0.5[a] | 0.3–0.9 | |
| Main effects | No. of visits | 1.7[b] | 1.1–2.4 | <0.01 |
| | Visitor[c] | | | <0.05 |
| *Period 2* | | | | |
| Main effects | GBS | 0.7[a] | 0.5–1.0 | <0.05 |
| | No. of visit | 1.4[b] | 1.0–1.8 | <0.05 |
| *Period 3* | | | | |
| Main effects | No. of visits | 1.7[b] | 1.2–2.4 | <0.01 |
| **Talk directed at the dog** | | | | |
| *Period 1* | | OR | 95% CI | P-value |
| Interaction | | | | |
| GBS * visit type | D | 0.8[a] | 0.4–1.6 | <0.05 |
| | DA | 0.2[a] | 0.1–0.6 | |
| Main effects | No. of visits | 1.6[b] | 1.1–2.3 | <0.05 |
| *Period 2* | | | | |
| Main effects | Speech (normal vs impaired) | 9.3 | 2.5–35.4 | 0.001 |
| | Handicap (normal vs impaired) | 7.7 | 1.8–33.4 | <0.01 |
| *Period 3* | | | | |
| Interaction | | | | |
| GBS * visit type | D | 1.0[a] | 0.5–1.8 | <0.05 |
| | DA | 0.2[a] | 0.1–0.7 | |
| Main effects | No. of visits | 2.1[b] | 1.3–3.2 | <0.01 |
| | Speech (normal vs impaired) | 5.1 | 1.0–26.2 | 0.05 |
| **Talking about the dog** | | | | |
| *Period 1* | | OR | 95% CI | P-value |
| Main effects | Vision (normal vs impaired) | 0.3 | 0.1–0.9 | <0.05 |
| | Visitor[c] | | | <0.05 |
| *Period 2* | | | | |
| Main effects | Visit type (D vs DA) | 0.4[b] | 0.1–1.0 | 0.05 |
| | Vision (normal vs impaired) | 0.3 | 0.1–0.9 | <0.05 |
| *Period 3* | | | | |
| Interaction | | | | |
| GBS * visit type | D | 1.5[a] | 0.9–2.4 | <0.01 |
| | DA | 0.5[a] | 0.3–1.0 | |
| Main effects | Vision (normal vs impaired) | 0.3 | 0.1–0.9 | <0.05 |

For each of the three periods, the estimated odds ratios (OR), the 95 confidence intervals (95% CI) and the P-values are shown for significant effects. Visits with a dog, no activity (D), Visits with a dog, with activity (DA), The Gottfries-Bråne-Steen Scale (GBS).

[a] For an increase of 20 on the GBS scale,

[b] For an increase of one visit,

[c] Difference between individual visitors not shown.

the cognitively well-functioning residents talked more about the dog in DA compared to D, and oppositely for the severely impaired participants. Throughout the intervention,

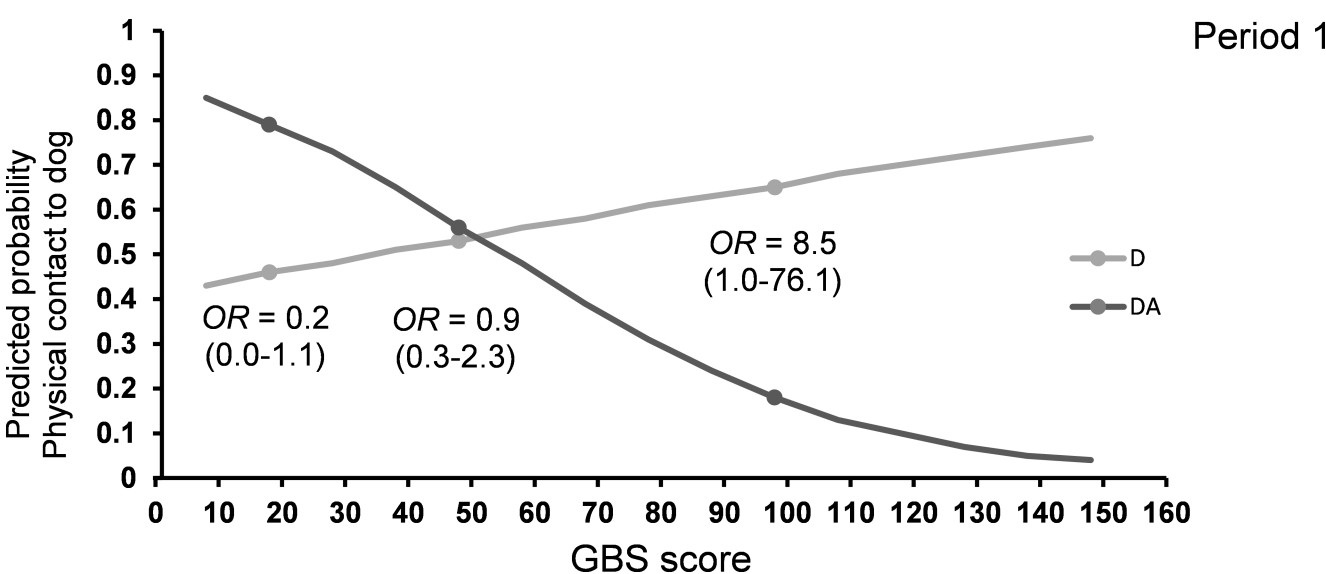

**Fig 4. Predicted probability for *physical contact with the dog* in period 1 estimated for the range of GBS scores and separately for D and DA visits.** D: visits with a dog, no activity, DA: visits with a dog, with activity. GBS scores of 18, 48 and 98 are marked with a dot. The odds ratios (OR) for each of these GBS-values are shown. The Gottfries-Bråne-Steen Scale (GBS).

residents with impaired vision were more likely to *talk about the dog* (period 1: $F_{1,81} = 4.81$, $P < 0.05$; period 2: $F_{1,81} = 5.15$, $P < 0.05$; period 3: $F_{1,84} = 5.28$, $P < 0.05$; Table 5). The identity of the visiting person only affected the talking about the dog in the first third of the intervention ($F_{5,81} = 2.43$, $P < 0.05$).

We found no effects of the residents' age and sex on their behaviour related to the dog in any of the periods. Over the three periods (12 visits), the residents' likelihood to touch the dog increased (Table 6), whereas the probability of talking to and about the dog decreased (Table 6).

**Initial 2 minutes of the visit.** In the first two minutes of the visits, the likelihood of touching the dog was lower in highly impaired residents in the first and last third of the intervention (period 1: $F_{1,92} = 9.69$, $P < 0.01$; period 3: $F_{1,89} = 6.78$, $P = 0.01$; Table 7), and, in the last two thirds, a higher number of total visits increased the odds (period 2: $F_{1,89} = 6.14$, $P < 0.05$; period 3: $F_{1,89} = 4.11$, $P < 0.05$; Table 7). Furthermore, higher age increased the odds of touching the dog during the first two minutes of a visit in period 2 ($F_{1,89} = 5.31$, $P < 0.05$; Table 7).

The likelihood of *talk directed at the dog* in the first two minutes decreased with increasing impairment level in period 1 and 2 of the intervention (period 1, $F_{1,90} = 9.51$, $P < 0.01$; period 2, $F_{1,92} = 6.39$, $P = 0.01$; Table 7). In the first period, residents with no physical handicap were more likely to talk to the dog compared to residents having a physical handicap ($F_{1,90} = 7.29$, $P < 0.01$; Table 7). In the last third of the intervention, talking directly to the dog was affected by visit treatment and GBS score in interaction ($F_{1,89} = 4.13$, $P < 0.05$), and residents with severe cognitive impairment talked less to the dog in DA compared to D visits (Table 7).

In the first third of the intervention, residents without physical handicaps and being able to talk were five times more likely to talk about the dog compared to residents with these impairments ($F_{1,89} = 5.65$, $P < 0.05$ and $F_{1,89} = 5.57$, $P < 0.05$; Table 7). In period 2, those who ended up having more visits were more likely to talk about the dog in the first two minutes ($F_{1,91} =$

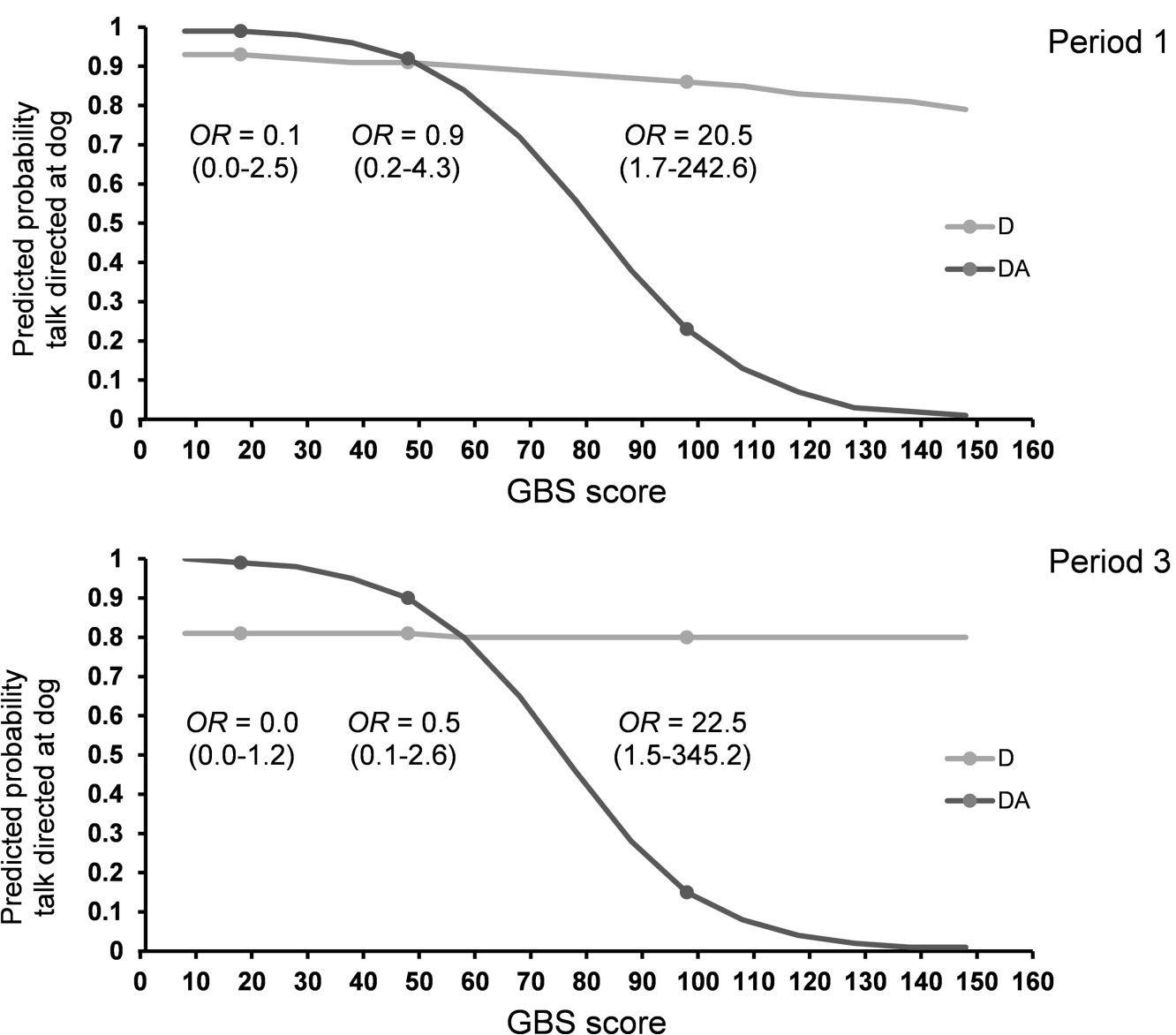

**Fig 5. Predicted probability for *talking directly to the dog* in periods 1 and 3 estimated for the range of GBS scores and separately for D and DA visits.** D: visits with a dog, no activity, DA: visits with a dog, with activity. GBS scores of 18, 48 and 98 are marked with a dot. The odds ratios (OR) for each of these GBS values are shown. The Gottfries-Bråne-Steen Scale (GBS).

5.84, P < 0.05; Table 7), and, in the last period, being 10 years older doubled the odds of talking about the dog ($F_{1,89}$ = 5.96, P < 0.05; Table 7).

### Behaviour recorded in visit types DA and A: *Physical contact to activity* and *Talking about the activity*

**Entire visit.**   Physical contact to the objects related to the activity (*physical contact to activity*) was affected by visit type in the last period of the intervention and tended to be so in the first (period 1: $F_{1,89}$ = 3.26, P = 0.07; period 3: $F_{1,92}$ = 14.44, P < 0.001; Table 8). The odds of taking part in the activity, in the last third of the intervention were 12.5 times higher for residents having visits without dogs (A) compared to those having DA visits. Throughout, the

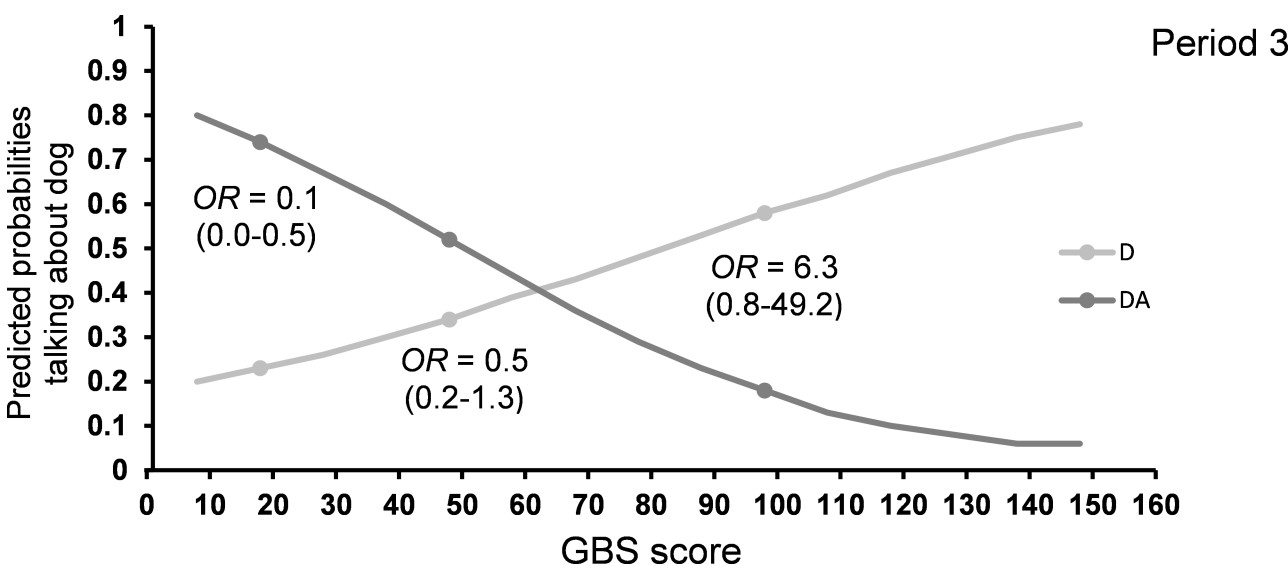

**Fig 6. Predicted probability for *talking about the dog* in period 3 estimated for the range of GBS scores and separately for D and DA visits.** D: visits with a dog, no activity, DA: visits with a dog, with activity. GBS scores of 18, 48 and 98 are marked with a dot. The odds ratios (OR) for each of these GBS values are shown. The Gottfries-Bråne-Steen Scale (GBS).

higher the GBS score, the lower the odds of taking part in the activity (period 1: $F_{1,89}$ = 7.89, P < 0.01; period 2: $F_{1,97}$ = 6.76, P = 0.01; period 3: $F_{1,92}$ = 13.57, P < 0.001; Table 8), showing that more impaired residents participated less in the activities. In the first third of the intervention period, residents with impaired vision touched the objects more compared to the rest of the population ($F_{1,89}$ = 6.07; P < 0.05; Table 8). Finally, ending up having many visits in total resulted in an increased probability of taking part in the activity in the last period of the intervention ($F_{1,92}$ = 12.05; P = 0.001; Table 8).

The probability of talking about the activity, including talking about the dog in relation to the activity in DA, was higher in A visits throughout the intervention (period 1: $F_{1,89}$ = 9.11, P = 0.01; period 2: $F_{1,88}$ = 12.81, P < 0.001; period 3: $F_{1,86}$ = 12.96, P < 0.001; Table 8).

**Table 6. The estimated probabilities of behaviour related to either the dog (D and DA visits) or the activity (DA and A visits).**

| Behaviour variable | Period | | | Statistics |
|---|---|---|---|---|
| | **1** | **2** | **3** | |
| Physical contact to dog (D, DA) | 0.37[c] [0.23; 0.59] | 0.49[b] [0.33; 0.58] | 0.52[a] [0.35; 0.74] | 1 vs 2: n = 95, S = 555 P < 0.05 |
| | | | | 1 vs 3: n = 94 S = 1008 P < 0.001 |
| | | | | 2 vs 3: n = 94, S = 670 P < 0.01 |
| Talk directed at the dog (D, DA) | 0.92[a] [0.84; 0.95] | 0.90[ab] [0.73; 0.95] | 0.92[b] [0.68; 0.96] | 1 vs 3: n = 94, S = -546 P < 0.05 |
| Talk about the dog (D, DA) | 0.25[b] [0.13; 0.50] | 0.31[a] [0.17; 0.55] | 0.34[ab] [0.19; 0.48] | 1 vs.. 2: n = 90, S = 857 P < 0.001 |
| Physical contact to activity (DA, A) | 0.60[b] [0.38; 0.75] | 0.61[a] [0.49; 0.72] | 0.70[a] [0.36; 0.90] | 1 vs 2: n = 93, S = 775 P < 0.01 |
| | | | | 1 vs 3: n = 91, S = 543 P < 0.05 |
| Talking about activity (DA, A) | 0.57[b] [0.25; 0.81] | 0.84[a] [0.45; 0.94] | 0.72[a] [0.43; 0.92] | 1 vs 2: n = 93 S = 1801 P < 0.0001 |
| | | | | 1 vs 3: n = 88, S = 793 P < 0.001 |

The estimated probabilities are shown as median and interquartile range (IQR) for each period. The estimates derive from the final generalised linear mixed models, and are corrected for significant factors in the respective models. Letters within variable (a, b) indicate differences across period. Visits with a dog, no activity (D); Visits with a dog, with activity (DA); Visits without dog, with an activity (A).

**Table 7. Results from the analysis of behaviour related to the dog in the first two minutes of D and DA visits.**

| Physical contact to the dog | | | | |
|---|---|---|---|---|
| *Period 1* | | *OR* | *95% CI* | *P-value* |
| Main effects | GBS | 0.5[a] | 0.3–0.8 | <0.01 |
| *Period 2* | | | | |
| Main effects | No. of visit | 1.5[b] | 1.1–2.0 | <0.05 |
| | Age | 2.1[c] | 1.1–4.2 | <0.05 |
| *Period 3* | | | | |
| Main effects | GBS | 0.6[a] | 0.4–0.9 | = 0.01 |
| | No. of visits | 1.4[b] | 1.2–4.2 | <0.05 |
| **Talk directed at the dog** | | | | |
| *Period 1* | | *OR* | *95% CI* | *P-value* |
| Main effects | GBS | 0.5[a] | 0.4–0.8 | <0.01 |
| | Handicap (normal vs impaired) | 6.1 | 1.6–23.0 | <0.01 |
| *Period 2* | | | | |
| Main effects | GBS | 0.6[a] | 0.5–0.9 | = 0.01 |
| *Period 3* | | | | |
| Interaction | D | 0.9[a] | 0.6–1.4 | <0.05 |
| GBS * visit type | DA | 0.4[a] | 0.2–0.8 | |
| **Talking about the dog** | | | | |
| *Period 1* | | *OR* | *95% CI* | *P-value* |
| Main effects | Handicap (normal vs impaired) | 5.4 | 1.3–21.7 | <0.05 |
| | Speech (normal vs impaired) | 5.2 | 1.3–20.5 | <0.05 |
| *Period 2* | | | | |
| Main effects | No. of visit | 1.4[b] | 1.1–1.8 | <0.05 |
| *Period 3* | | | | |
| Main effects | Age | 2.0[c] | 1.1–3.4 | <0.05 |

For each of the three periods, the estimated odds ratios (OR), the 95 confidence intervals (95% CI), and the P-values are shown for significant effects. Visits with a dog, no activity (D), Visits with a dog, with activity (DA), The Gottfries-Bråne-Steen Scale (GBS).

[a] For an increase of 20 on the GBS scale,

[b] For an increase of 1 visit,

[c] For an increase of 10 years.

Generally, the odds of talking about the activity were higher, the lower the GBS score (period 1: $F_{1,89} = 13.78$, P <0.001; period 2: $F_{1,88} = 3.27$, P = 0.07; period 3: $F_{1,86} = 3.88$, P = 0.05; Table 8), meaning that the more impaired residents talked less about the activity during the visits. In the first third of the intervention, talking about the activity was related to having many visits in total, the odds increasing with 1.6 for each extra visit ($F_{1,92} = 5.06$, P < 0.05; Table 8). Being hearing-impaired, as opposed to having good hearing, increased the odds of talking about the activity throughout (period 1: $F_{1,89} = 12.11$, P < 0.001; period 2: $F_{1,88} = 7.24$, P < 0.01; period 3: $F_{1,86} = 6.77$, P = 0.01; Table 8). Residents with impaired speech were less likely to talk about the activity in the last two thirds of the intervention (period 2: $F_{1,88} = 6.84$, P = 0.01 period 3: $F_{1,86} = 6.77$, P = 0.01; Table 8).

We found no effects of the residents' age and sex on their behaviour related to the activity in any of the periods, and the probability of having physical contact to and talking about the activity increased over time (Table 6).

**Table 8. Results from the analysis of behaviour related to an activity in DA and A visits.**

| Physical contact to activity | | | | |
|---|---|---|---|---|
| *Period 1* | | *OR* | *95% CI* | *P-value* |
| Main effects | Visit type (A vs DA) | 2.4 | 0.9–6.0 | 0.07 |
| | GBS | 0.6[a] | 0.4–0.8 | <0.01 |
| | Vision (normal vs impaired) | 0.3 | 0.1–0.8 | <0.05 |
| *Period 2* | | | | |
| Main effects | GBS | 0.6[a] | 0.5–0.9 | 0.01 |
| *Period 3* | | | | |
| Main effects | Visit type (A vs DA) | 12.5 | 3.3–45.5 | <0.001 |
| | GBS | 0.4[a] | 0.2–0.7 | <0.001 |
| | No. of visits | 2.1[b] | 1.4–3.1 | <0.001 |
| **Talking about activity** | | | | |
| *Period 1* | | *OR* | *95% CI* | *P-value* |
| Main effects | Visit type (A vs DA) | 6.0 | 1.8–19.2 | <0.01 |
| | GBS | 0.4[a] | 0.2–0.6 | <0.001 |
| | No. of visits | 1.6[b] | 1.1–2.5 | <0.05 |
| | Hearing (normal vs impaired) | 0.1 | 0.0–0.4 | <0.001 |
| *Period 2* | | | | |
| Main effects | Visit type (A vs DA) | 11.1 | 2.9–41.7 | <0.001 |
| | GBS | 0.6[a] | 0.3–1.1 | 0.07 |
| | Hearing (normal vs impaired) | 0.1 | 0.0–0.6 | <0.01 |
| | Speech (normal vs impaired) | 9.1 | 1.7–49.1 | 0.01 |
| *Period 3* | | | | |
| Main effects | Visit type (A vs DA) | 10.6 | 2.9–38.5 | <0.001 |
| | GBS | 0.6[a] | 0.3–1.0 | 0.05 |
| | Hearing (normal vs impaired) | 0.2 | 0.1–0.9 | 0.01 |
| | Speech (normal vs impaired) | 9.9 | 1.7–57.5 | 0.01 |

For each of the three periods, the estimated odds ratios (OR), the 95 confidence intervals (95% CI), and the P-values are shown for significant effects. Visits with a dog, with activity (DA), Visits without dog, with an activity (A). The Gottfries-Bråne-Steen Scale (GBS).

[a] For an increase of 20 on the GBS scale,

[b] For an increase of 1 visit.

## Discussion

Visit type, but also the residents' level of cognitive impairment, affected their behaviour during the visits. In general, adding an activity to a visit resulted in less interaction with the dog for the residents with a high impairment level compared to dog visits without an activity. On the contrary, for less impaired residents, interacting with the dog was either unaffected or more likely by adding an activity to dog visits. In visits including an activity, the residents' engagement, measured as touching the objects related to the activity and talking about it, was less likely when a dog was present compared to visits without a dog. The duration of talking to the persons present during visits was shorter the higher the impairment level.

We expected that introducing the option of engaging in an activity with the dog during dog visits would stimulate the residents to interact more with the dog and that especially the least impaired residents would be motivated/stimulated by this increased complexity and take more actively part. Our findings are partly in support of this. The residents' impairment level indeed

affected how much they interacted during the two types of dog visits. However, not all residents reacted to the inclusion of an activity with increased dog interaction. In the first period of the intervention, the residents touched the visiting dog less with increasing impairment level, if the visit included an activity with the dog. Throughout the intervention, the most impaired residents talked less to and about the dog when they received dog visits with activities compared to without activities. The pattern was different for the more cognitively intact residents. They were either unaffected by visit type or more active in DA visits, and for example in the last third of the intervention period, they talked more about the dog in DA visits compared to the D visits. Overall, physical contact to the dog and the activity, as well as talking about both, increased over time, whereas talking directly to the dog decreased in the last third of the intervention period.

Based on these results, the overall impression is that the extra stimulation, by repeatedly introducing a new activity during dog visits, stimulated the more cognitively well-functioning participants to interact more, whereas the severely impaired residents did not seem to benefit from this increased complexity. Physical contact with the dog did not differ according to visit type and impairment level in the last third of the intervention period, which could mean that the engagement in the visits also depended on the familiarity with the concept of the visits, including the dog and persons.

Some previous studies of dog visits in nursing homes have also included observations of nursing home residents' immediate behavioural responses; however, mostly during group interventions, employing less detailed behaviour sampling methods and addressing different research questions [19–22, 34–37]. Some of these found that dog visits, as opposed to visits without dogs, stimulate the residents to interact more with other persons during group sessions [19, 21, 37], whereas Kramer et al. (2009) [34] did not find this effect.

Our previous findings, i.e. that dog visits were more stimulating and had a longer-lasting effect, compared to visits with a toy cat, and the fact that severely impaired compared to less impaired residents interacted more with the dog and less with persons present during visits [18], are partly supported by the present study. At present, we found a negative correlation between increased impairment level and talking to persons, regardless of topic. However, in D visits that were comparable to the dog visits in Thodberg et al. (2016), we only found a negative correlation between impairment level and the probability of touching the dog in the first third of the intervention period, but not later on. Apart from our present and previous study (Thodberg et al., 2016), we are only aware of two other studies that relate nursing home residents' cognitive ability to their behaviour during human-dog interactions–both carried out in group settings. Olsen et al. (2019) found that increased cognitive impairment had a negative effect on how much the residents talked during group visits with dogs, whereas Friedmann et al. (2019) did not find any variation in behaviour related to cognitive impairment. The existing studies, including this one, show that dog visits stimulate to conversation but that increasing impairment level results in less verbal communication with persons and in some cases more interaction with a dog–the latter depending on the complexity and how intellectually challenging the visits are. In Thodberg et al. (2016), we suggested that it may be more comfortable for residents with a lowered cognitive ability (which can result in speech impairment [38]) to interact with a non-evaluative animal as opposed to a person, as interaction with an animal does not necessarily require verbal communication [39]. The present study showed that being presented with an activity during dog visits resulted in reduced interaction with the dog the higher the residents' impairment level. This implies that visits with increased complexity might lower the motivation to interact with the dog. Perhaps the increased complexity, which seems to be a positive supplement for the less impaired residents, makes the visits too stimulating or overwhelming for the severely impaired residents.

In order to be able to distinguish between the effects of access to dog contact and the option to take part in an activity, we included visit type A without a dog but with the same repeated introduction to a new activity at each visit as in DA visits. When no dog was present, the residents talked more about the activity, and, in the last part of the intervention, they were also more engaged in the activity compared to in visits with both a dog and an activity. Furthermore, the residents touched the items related to the activity less and talked less about the ongoing activity with increasing cognitive function in DA visits. This demonstrates that the participants were generally less focussed on the activity when a dog was also part of the visits––especially residents with severe dementia–which might be due to them dividing their attention between the activity and the dog or them being distracted. However, it cannot be ruled out that difference in the type of activities in DA and A visits could have influenced the residents' motivation to join. Many of the activities chosen for A visits revolved around items and reminiscences from the past, whereas the activities in DA visits aimed at activating the residents to interact with the dog, often involving handling toys and treats for the dog. Talking about old times and things from the past could be considered more interesting by some residents, and an additional treatment group with the same items and presence of a dog would be necessary to investigate this further.

Furthermore, we observed a generally lowered engagement level in residents with severe dementia during visits including an activity, regardless of there being a dog present or not.

Our results are in line with findings of less attendance of residents with increasing impairment level in the activities normally available for nursing home residents [24–26, 40]. Furthermore, nursing home residents engage less in the actual activity with increasing impairment level–both in individual activities [41] and group activities [42, 43]. Activities presumed to be meaningful for the individual increase engagement, regardless of cognitive function, whereas work-related activities are more engaging for residents with moderate compared to severe dementia [41]. This shows that the type and content of the activity is indeed important and should be appropriate with regard to the cognitive as well as physical abilities of the individual [44, 45]. With the results from the present study, and general knowledge about participation in nursing home activities, the next step is to tailor dog visits, taking into account the different needs and abilities of residents with varying cognitive impairment levels.

The residents' response to visits depended on whether or not they had a physical handicap, and, as expected, participants without a speech impairment talked more about activities and the dog. Impaired vision resulted in a higher prevalence of talking about the dog throughout the intervention and increased odds of touching the objects during visits with an activity, whereas reduced ability to hear increased the likelihood of talking about the activity. It appears logical that visual impairment stimulates the residents to touch objects more and talk more about the dog in order to orient themselves and get acquainted with the situation and that reduced hearing could result in more verbal communication–maybe due to more questions and repetitions. The occurrence of physical contact to the dog was not affected by any of the handicaps, which might indicate that none of the handicaps hindered touching the dog. We are not aware of other studies considering effects of physical handicaps on immediate responses to dog visits. Cohen-Mansfield et al. (2009) [46] found no influence of impairment level in either hearing or eyesight on the engagement in different nursing home activities, but they noticed that residents with poor hearing are more likely to refuse participation in activities. Future studies should take into account that sensory deficits and other physical handicaps might influence the response to visits and, as it is the case for the level of cognitive impairment, should be controlled for in the analyses of outcome measures.

This type of research, where the experimental setup involves interactions with humans as well as animals, entails certain challenges regarding standardisation and hindering that the

response of the study participants depends on actions of the person present. We are very aware of these pitfalls, and therefore standardised the experimental visits by dividing them into three phases as described in the methods. Thus, we waited two minutes into the visit before introducing the activity in DA and A visits and only incited the residents to take part in the following four minutes, leaving the initiative to the residents in the remainder of the time. Apart from adding a clear structure to the visits, this approach enabled us to compare how residents, having visits with or without an activity, interacted with the dog in these identical two minutes of the visits, and whether their behaviour in these initial minutes changed over the intervention period; because some expected an activity (DA) whereas others did not (D). In the last third of the intervention period, we saw that the more severely impaired talked less to the dog if they received DA compared to D visits, not only in the visits as a whole but also in these first identical two minutes of the visits. The fact that they had the same response, even before being presented with the activity, suggests that their previous experience of being repeatedly presented with new activities reduced their motivation to interact with the dog late in the intervention period. This is especially interesting, keeping in mind that D and DA visits were identical in the initial two minutes, and that talking to the dog in visits without activities (D) in all 10 minutes was unaffected by cognitive impairment level in period 3.

Limitations to the study were that neither the participants nor the observers and visitors were blind regarding visit type, which is another challenge to this research field. However, to accommodate this, observer-visitor teams were not fixed. The two observers in each nursing home joined the two visitors an equal number of times to prevent the development of team-specific variation and drifting in conductance and observation of visits. Furthermore, both the identity of the observer and the visitor was included in the statistical model as fixed effects to be able to control for any variation related to them. We found no observer effect in any of the statistical analyses, but the identity of the visitor affected the probability of touching and talking about the dog in the first period of the intervention, which could be due to variation in visitors' personality and management/handling of the dog. Finally, we acknowledge that unaccounted-for variation in the participants' attitude to dogs/animals and the dynamics between two sentient beings (human and animal) could have influenced our results.

Summing up, the residents' immediate behavioural response indicates that standard dog visits with no additional activity, stimulate nursing home residents to interact physically and verbally with the dog, especially residents with severe dementia. Introducing an activity stimulated the more cognitively well-functioning residents to interact more but had the opposite effects on severely impaired residents–especially in the first part of a 6-week intervention period. The presence of a dog during visits with activities resulted in less engagement in the activity, and engagement in activities was generally lowered with increasing impairment level.

Based on the present knowledge, the optimal dog visit for the less cognitively impaired nursing home residents could include activities and thereby a possibility to interact in different ways with the dog, whereas for residents with severe impairment just being with the dog seems more appropriate.

## Supporting information

**S1 File. CONSORT checklist.**
(DOC)

**S2 File. Study protocol.**
(PDF)

## Acknowledgments

The authors wish to thank the participating residents and the staff in the participating nursing homes. We also thank the dog owners for lending us their dogs, and TrygFonden for testing the dogs and mediating the contact to the dog owners. We acknowledge the help from the nurses Ingrid Keseler, Leni Karkov Meeder and Birgitte Rasmussen (Aarhus University Hospital, Denmark) and Pia Haun Poulsen and Birthe Houbak (technical staff at Aarhus University, Denmark) and the students Johanne Foverskov Stonor Krum-Møller and Line Kollerup Oftedal and Kristina Høyby Simonsen for the role as visitors and observers. Finally, we acknowledge Leslie Foldager for giving advice regarding the statistical analyses.

## Author Contributions

**Conceptualization:** Karen Thodberg, Poul B. Videbech, Tia G. B. Hansen, Janne W. Christensen.

**Data curation:** Karen Thodberg, Anne Bak Pedersen.

**Formal analysis:** Karen Thodberg.

**Funding acquisition:** Karen Thodberg, Poul B. Videbech, Tia G. B. Hansen, Janne W. Christensen.

**Investigation:** Karen Thodberg, Anne Bak Pedersen, Janne W. Christensen.

**Methodology:** Karen Thodberg, Poul B. Videbech, Tia G. B. Hansen, Anne Bak Pedersen, Janne W. Christensen.

**Project administration:** Karen Thodberg.

**Resources:** Karen Thodberg.

**Supervision:** Karen Thodberg.

**Visualization:** Karen Thodberg.

**Writing – original draft:** Karen Thodberg.

**Writing – review & editing:** Karen Thodberg, Poul B. Videbech, Tia G. B. Hansen, Anne Bak Pedersen, Janne W. Christensen.

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
