## [Decision Letter · Decision Letter 0]

4 Feb 2021

PONE-D-20-34638

Dog visits in nursing homes – increase complexity or keep it simple? A randomised controlled study

PLOS ONE

Dear Dr. Thodberg,

Thank you for submitting your manuscript to PLOS ONE. After careful consideration, we feel that it has merit but does not fully meet PLOS ONE’s publication criteria as it currently stands. Therefore, we invite you to submit a revised version of the manuscript that addresses the points raised during the review process.

Comments of the external reviewers are listed below. Please would you give especial consideration to discussing the limitations of your study and its practical relevance in managing residents in nursing homes. I think the text is clear and easily understood throughout, but nevertheless you might consider asking a native English speaker to polish it.

We look forward to receiving your revised manuscript.

Kind regards,

Antony Bayer

Academic Editor

PLOS ONE

Journal Requirements:

Reviewers' comments:

Reviewer's Responses to Questions

**Comments to the Author**

1. Is the manuscript technically sound, and do the data support the conclusions?

Reviewer #1: Yes

Reviewer #2: Partly

Reviewer #3: Yes

2. Has the statistical analysis been performed appropriately and rigorously? 

Reviewer #1: I Don't Know

Reviewer #2: I Don't Know

Reviewer #3: Yes

3. Have the authors made all data underlying the findings in their manuscript fully available?

Reviewer #1: Yes

Reviewer #2: Yes

Reviewer #3: Yes

4. Is the manuscript presented in an intelligible fashion and written in standard English?

Reviewer #1: Yes

Reviewer #2: Yes

Reviewer #3: Yes

5. Review Comments to the Author

Reviewer #1: -This is a very well-written paper and undertaken, detailed study.

-In fact, the paper seems very much detailed at times, and may even seem repetitive. I would not suggest changing it in response to this comment, but I would suggest reading it through with this in mind in case there are places it can be streamlined.

-Lines 153-155 (; balancing). Awkward and unclear sentence.

-Throughout the paper could benefit from a read through for English grammar use.

-It would be helpful to share upfront that these visiting dogs are from an organization that tests dogs for such visits.

-Related, I am not clear if this organization also trains dogs? I am curious because the dogs are being involved in several activities (e.g., doing tricks with the dice roll game). Also, the dogs must be quite well trained to go off with a stranger and not their regular handler to do a visit of this type. I think this needs to be identified clearly for the reader.

-And related again, this raises the question of animal welfare and more specific information needs to be included. Was an animal research ethics board application undertaken? For example, if the dogs was standing or near the individual for 80% of the time I am assuming they were trained to do this and not physically forced to do so because of a short leash, for example. I am also assuming they are very well trained because seniors are typically accompanied with a lot of bodily and medication scents of dogs’ interest and this did not get mentioned in terms of how the dogs interacted with the individuals.

-Overall, I would like to see some attention, if even in the limitations, acknowledging that animals are not tools but sentient beings. So ‘something’ may be going on between the participant and dog that is not being accounted for here; the human-animal bond. And it is fine that it is not being accounted for, but this needs to be shared with the reader.

-And last, I think more attention needs to be placed on who the individuals were doing the visits. I find it interesting and would like to know how they worked with the dogs in the visits in place of the handler. Visiting dog teams typically come together as a unit, so what impact did separating this visiting unit have, if any?

Reviewer #2: Important note: This review pertains only to ‘statistical aspects’ of the study and so ‘clinical aspects’ [like medical importance, relevance of the study, ‘clinical significance and implication(s)’ of the whole study, etc.] are to be evaluated [should be assessed] separately/independently. Further please note that any ‘statistical review’ is generally done under the assumption that (such) study specific methodological [as well as execution] issues are perfectly taken care of by the investigator(s). This review is not an exception to that and so does not cover clinical aspects {however, seldom comments are made only if those issues are intimately / scientifically related & intermingle with ‘statistical aspects’ of the study}. Agreed that ‘statistical methods’ are used as just tools here, however, they are vital part of methodology [and so should be given due importance].

COMMENTS: Actually, your ABSTRACT is well drafted but assay type. Please note that it is preferable [refer to item 1b of CONSORT checklist 2010: Structured summary of trial design, methods, results, and conclusions] to divide the ABSTRACT with small sections like ‘Objective(s)’, ‘Methods’, ‘Results’, ‘Conclusions’, etc. which is an accepted practice of most of the good/standard journals [including this one]. It will definitely be more informative then, I guess, whatever the article type may be.

I agree with what Friedman et al. (2019) reported according to authors of this article that [lines 97-98] ‘no association between impairment level, and recorded behaviour during dog visits’. Important factor not taken care of here is ‘whether subject [dementia case] like the pets?’ In my experience, there are many people who do not like pets {I am not very sure about situation in Europe (though I had been there several times and stayed), but I am very much sure about situation in my country}. This factor could be a ‘confounder’ and needs to be adjusted for, [or may totally exclude such cases], in my opinion.

Because of the above and authors aimed to study whether enhancing the possibility for interaction with the dog during dog visits is beneficial for all [as said in lines 109-10], this review may be ‘biased’, I think. Although one of the exclusion criteria {line 144} is ‘fear of dogs’, I guess, not liking pets/dogs is not exactly same as ‘fear of dogs’. Mention in line 194/195 of the fact that ‘disapproved of contact with the animal’ implies there were at least few of this type.

Information given regarding estimation of ‘Sample size required’ (lines 164-6) is inadequate. Is it assumed that 12 (each) different activities listed in table 1 & 2 are of the same type/intensity? Is it correct to impute/substitute the values of the missed items with the median [of this item across participants]? In back-drop of ‘level of measurement’ {and/or type of variables} of ‘Behaviour variables’ [all measurements], ‘Non-parametric models’ are more sensible it seems. Many other things are analysed and discussed. Most of them appears alright [statistically], however, ‘clinical aspects’ [like medical importance, relevance of the study, ‘clinical significance and implication(s)’ of the whole study, etc.] are to be evaluated [should be assessed] separately/independently, I feel.

Reviewer #3: The manuscript is clear in every part. The description of the activities carried out in the meetings with the patients allows to better specify the results, which represented an interesting starting point for further development of studies on this topic

6. PLOS authors have the option to publish the peer review history of their article (what does this mean?). If published, this will include your full peer review and any attached files.

Reviewer #1: No

Reviewer #2: No

Reviewer #3: No

---

## [Author Response · Author response to Decision Letter 0]

25 Mar 2021

PONE-D-20-34638

Dog visits in nursing homes – increase complexity or keep it simple? A randomised controlled study

PLOS ONE

We have copied in the review questions and written the answers below each question

Reviewer #1: -This is a very well-written paper and undertaken, detailed study.

-In fact, the paper seems very much detailed at times, and may even seem repetitive. I would not suggest changing it in response to this comment, but I would suggest reading it through with this in mind in case there are places it can be streamlined.

Answer: Thank you for this comment. We have read the manuscript carefully and shortened the text where possible (Lines: 84-85, 174-175, 202, 207-208).

-Lines 153-155 (; balancing). Awkward and unclear sentence.

Answer: This has been rephrased (now line 166-168).

-Throughout the paper could benefit from a read through for English grammar use.

Answer: The manuscript has been checked for grammar now and small adjustments have been made throughout the manuscript.

-It would be helpful to share upfront that these visiting dogs are from an organization that tests dogs for such visits.

Answer: I have moved the sentence describing the organization to the start of the section about the dogs included in the study (now lines 242-246). We agree that this information is important and that it should be clearly stated that the dogs lived up to certain standards, both to ensure their welfare and to standardize the study as much as possible. 

-Related, I am not clear if this organization also trains dogs? I am curious because the dogs are being involved in several activities (e.g., doing tricks with the dice roll game). Also, the dogs must be quite well trained to go off with a stranger and not their regular handler to do a visit of this type. I think this needs to be identified clearly for the reader.

Answer: The organization (TrygFonden besoegshunde) did not train the dogs specifically for his task, but the dogs and their owners have to pass a test and a check by a veterinarian to be included in the corps of visiting dogs. This is to ensure that they were comfortable with the tasks involved during nursing home visits. (new place, being handled by different people, unknown noise etc). Before choosing the dogs that could be part of this specific project, the handler met with the dogs and their owners a number of times before the experimental activities started, to get to know each other and find out whether they were a good team. In the meetings, they also tried out the different activities. The activities were to some extent adapted to the individual dogs. For us it was not so important that the dogs could do the tasks perfectly, but that they were motivated to be part of the visits and relaxed with the handler. We have clarified this in the text (lines 250-252).

-And related again, this raises the question of animal welfare and more specific information needs to be included. Was an animal research ethics board application undertaken? For example, if the dogs was standing or near the individual for 80% of the time I am assuming they were trained to do this and not physically forced to do so because of a short leash, for example. I am also assuming they are very well trained because seniors are typically accompanied with a lot of bodily and medication scents of dogs’ interest and this did not get mentioned in terms of how the dogs interacted with the individuals.

Answer: Yes, the dogs were all well-trained and were never forced to stay in position. If they moved away from the resident they were signaled to go back and were positively reinforced for doing so (verbal praise, stroking and sometimes a treat). Indeed, the handlers were very much aware that they were safeguarding the dogs’ welfare. This has now been clarified (lines 253-254).

We did not apply for an ethical permit for the use of dogs in this study. According to the Danish law regarding animals in research, the threshold for when to seek permission is when animals are subjected to procedures comparable to a needle prick or worse, and the activities in this project were not considered to be in that category.

-Overall, I would like to see some attention, if even in the limitations, acknowledging that animals are not tools but sentient beings. So ‘something’ may be going on between the participant and dog that is not being accounted for here; the human-animal bond. And it is fine that it is not being accounted for, but this needs to be shared with the reader.

Answer: That is good point and thank for bringing this up. We have included this in the discussion about limitations (lines 758-760).

-And last, I think more attention needs to be placed on who the individuals were doing the visits. I find it interesting and would like to know how they worked with the dogs in the visits in place of the handler. Visiting dog teams typically come together as a unit, so what impact did separating this visiting unit have, if any?

Answer: As explained above, we aimed for the handler and dog to “team up” before the experimental period through visits in the dogs’ home, and the handler and dog remained “a team” throughout the visiting period. We generally aimed to standardize the visits, but handler-dog teams are of course different. Therefore, we ensured that each handler-dog team did an equal amount of the three visit types to balance out this variation. Furthermore, we controlled for any variation in the statistical analysis and it turned out that the visitor (handler) only affected the results in a few cases (see line 756-758). 

Reviewer #2: Important note: This review pertains only to ‘statistical aspects’ of the study and so ‘clinical aspects’ [like medical importance, relevance of the study, ‘clinical significance and implication(s)’ of the whole study, etc.] are to be evaluated [should be assessed] separately/independently. Further please note that any ‘statistical review’ is generally done under the assumption that (such) study specific methodological [as well as execution] issues are perfectly taken care of by the investigator(s). This review is not an exception to that and so does not cover clinical aspects {however, seldom comments are made only if those issues are intimately / scientifically related & intermingle with ‘statistical aspects’ of the study}. Agreed that ‘statistical methods’ are used as just tools here, however, they are vital part of methodology [and so should be given due importance].

COMMENTS: Actually, your ABSTRACT is well drafted but assay type. Please note that it is preferable [refer to item 1b of CONSORT checklist 2010: Structured summary of trial design, methods, results, and conclusions] to divide the ABSTRACT with small sections like ‘Objective(s)’, ‘Methods’, ‘Results’, ‘Conclusions’, etc. which is an accepted practice of most of the good/standard journals [including this one]. It will definitely be more informative then, I guess, whatever the article type may be.

Answer: The abstract has been re-structured accordingly.

I agree with what Friedman et al. (2019) reported according to authors of this article that [lines 97-98] ‘no association between impairment level, and recorded behaviour during dog visits’. 

Answer: We are not sure that we understand this comment. In our study, we found that impairment level affected the behaviour during dog visits, which is in contrast to the finding in Friedman et al (2019).

Important factor not taken care of here is ‘whether subject [dementia case] like the pets?’ In my experience, there are many people who do not like pets {I am not very sure about situation in Europe (though I had been there several times and stayed), but I am very much sure about situation in my country}. This factor could be a ‘confounder’ and needs to be adjusted for, [or may totally exclude such cases], in my opinion.

Answer: In a previous study on visiting dog in nursing homes, we tried to include questions about attitude and previous experience with pets, but found it very difficult to get valid information. Especially due to participants’ lack of memory. Therefore this was not included in the present study. Participation is the study was voluntary and the residents were informed that visits could include a dog. For cognitively impaired residents, a close relative gave informed consent. We doubt that anyone would sign up for a study on visiting dogs if they generally disliked dogs. In the discussion (lines 758-760), we have inserted a sentence discussing this limitation of the study.

Because of the above and authors aimed to study whether enhancing the possibility for interaction with the dog during dog visits is beneficial for all [as said in lines 109-10], this review may be ‘biased’, I think. Although one of the exclusion criteria {line 144} is ‘fear of dogs’, I guess, not liking pets/dogs is not exactly same as ‘fear of dogs’. Mention in line 194/195 of the fact that ‘disapproved of contact with the animal’ implies there were at least few of this type.

Answer: We agree that variation in how much the residents like pets may affect the results. We did not collect data related to this. In a previous study we tried this, for the same population (as mentioned above). 

Information given regarding estimation of ‘Sample size required’ (lines 164-6) is inadequate. 

Answer: We have clarified that the data used for power calculations were behavioural data from our previous studies (lines 176-178)

Is it assumed that 12 (each) different activities listed in table 1 & 2 are of the same type/intensity? 

Answer: The activities were chosen to activate different senses and abilities. We aimed, as far as possible, to select the same type of activities for visits with and without a dog present, but it was difficult to do completely comparable activities as also discussed in lines 692 – 699.

Is it correct to impute/substitute the values of the missed items with the median [of this item across participants]? 

Answer: Imputation by inserting the median value is a relative simple imputation method. The two variables, GBS and MMSE (both measures of cognitive impairment) were highly correlated, and we used GBS as our explanatory variable, as it had very few missing observations (3 out of the 151 participant (2%). We have added this information in lines 280-281.We believe that imputation by the median in this case is a valid choice. We have added a reference that compares different imputation methods and relates them to different proportions of missing observation (line 279).

In back-drop of ‘level of measurement’ {and/or type of variables} of ‘Behaviour variables’ [all measurements], ‘Non-parametric models’ are more sensible it seems. 

Answer: I am not sure whether this is a question?

Many other things are analysed and discussed. Most of them appears alright [statistically], however, ‘clinical aspects’ [like medical importance, relevance of the study, ‘clinical significance and implication(s)’ of the whole study, etc.] are to be evaluated [should be assessed] separately/independently, I feel.

Reviewer #3: The manuscript is clear in every part. The description of the activities carried out in the meetings with the patients allows to better specify the results, which represented an interesting starting point for further development of studies on this topic

Answer: Thank you �

In addition to the above answers and changes, we have made the following adjustments.

We have inserted the captions for two files with supplementary information.

- S1 is the CONSORT Checklist that have updated with the correct page numbers (referring to in the manuscript version with accepted track changes).

- S2 is the Study protocol.

We have a question regarding the study protocol. We prefer to keep it in its present form and not change it to the form in “protocols.io”, would that be ok?

In figure 1, we have inserted the abbreviation for the visit type in the boxes under Allocation and Analysis. 

There was one reference that were not correctly inserted (Martin and Bateson, no. 32) and this has been corrected. In a few places, where the reference was the grammatical subject in a sentence, the reference was refered to with the two first authors – we have changes this to for example “Thodberg et al. (2016) found….” With a reference to the paper in squared brackets. Is this correct?

---

## [Editor Report · Decision Letter 1]

6 Apr 2021

PONE-D-20-34638R1

Dog visits in nursing homes – increase complexity or keep it simple? A randomised controlled study

PLOS ONE

Dear Dr. Thodberg,

Thank you for submitting your manuscript to PLOS ONE and for your careful attention to addressing the reviewers' comments. An outstanding issue is your statement concerning sample size that remains inadequate. It should include all the specific numerical details that will allow the reader to reproduce your calculation - rather than the current broad description ("behavioural data from our previous studies").Therefore your paper does not fully meet PLOS ONE’s publication criteria as it currently stands and we invite you to submit a revised version of the manuscript that addresses the points raised.

We look forward to receiving your revised manuscript.

Kind regards,

Antony Bayer

Academic Editor

PLOS ONE
---

## [Author Response · Author response to Decision Letter 1]

23 Apr 2021

Question from the editor:

An outstanding issue is your statement concerning sample size that remains inadequate. It should include all the specific numerical details that will allow the reader to reproduce your calculation - rather than the current broad description ("behavioural data from our previous studies"

Answer:

I have revised the section about sample size and included more numerical details (lines: 163-173).

The line numbers refer to the manuscript version without track changes. 

In Table 1, the number five “5” has been added in the line describing activity number five.

The manuscript with track changes is the same version as submitted in March, and include the track changes made back then as well as the present changes

---

## [Editor Report · Decision Letter 2]

29 Apr 2021

Dog visits in nursing homes – increase complexity or keep it simple? A randomised controlled study

PONE-D-20-34638R2

Dear Dr. Thodberg,

Thank you for your most recent manuscript. The sample size calculation is now clear. Therefore we’re pleased to inform you that your manuscript has been judged scientifically suitable for publication and will be formally accepted for publication once it meets all outstanding technical requirements.

Kind regards,

Antony Bayer

Academic Editor

PLOS ONE
---

## [Editor Report · Acceptance letter]

3 May 2021

PONE-D-20-34638R2 

Dog visits in nursing homes – increase complexity or keep it simple? A randomised controlled study 

Dear Dr. Thodberg:

I'm pleased to inform you that your manuscript has been deemed suitable for publication in PLOS ONE. Congratulations! Your manuscript is now with our production department. 

Kind regards, 

on behalf of

Professor Antony Bayer 

Academic Editor

PLOS ONE